# Thin Films of Nonlinear Metallic Amorphous Composites

**DOI:** 10.3390/nano12193359

**Published:** 2022-09-27

**Authors:** Navid Daryakar, Christin David

**Affiliations:** 1Institute of Condensed Matter Theory and Optics, Friedrich-Schiller-Universität Jena, Max-Wien-Platz 1, 07743 Jena, Germany; 2Abbe Center of Photonics, Albert-Einstein-Straße 6, 07745 Jena, Germany

**Keywords:** nonlinear optics, nanoparticles, thin films, effective medium theory

## Abstract

We studied the nonlinear optical response of metallic amorphous composite layers in terms of a self-phase-modulated, third-order Kerr nonlinearity. A nonlinear effective medium theory was used to describe low densities of gold and iridium nanoparticles embedded in an equally nonlinear host material. The fill fraction strongly influences the effective nonlinear susceptibility of the materials, increasing it by orders of magnitude in the case of gold due to localized surface plasmonic resonances. The enhancement of the nonlinear strength in amorphous composites with respect to the bulk material has an upper limit in metallic composites as dominating absorption effects take over at higher fill factors. Both saturated and induced absorption in the thin films of amorphous composites were observed depending on the selected frequency and relative position to the resonant frequency of electron excitation in the metallic inclusions. We demonstrated the depths to which thin films are affected by nonlinear enhancement effects.

## 1. Introduction

Amorphous composite materials can be obtained from techniques such as atomic layer deposition (ALD) with a high precision with respect to the final film thickness, composite density *f* and surface morphology, with a very low root mean square (rms) randomness of the obtained surfaces [1,2]. If using metal inclusions, such amorphous layers can be used for increasing local fields inside the host matrix, allowing for intense near-field interactions with the host material, e.g., in polymer platforms for photovoltaic devices such as dye-sensitized and other organic solar cell technologies [3,4]. On the other hand, should the composite materials exhibit nonlinear optical properties, the total nonlinear output fields can likewise be enhanced due to the presence of nanoparticles. Ultimately, the design of nonlinear metasurfaces [5,6,7,8] has the potential to add a new dimension to the control of light fields and enhancement phenomena at the nanoscale.

Amorphous composites of low fill fraction *f* both in solids and in liquids are typically studied within Maxwell–Garnett (M-G) [9] and Bruggeman (Br) theories and, according to both of them, nonlinear extensions exist [10,11,12]. The quality of effective medium theories [13,14,15] is strongly limited by the approximations used, such as the quasistatic, non-interacting regime, omitting multiple scattering and coupling effects. In this work, we studied the thin films of amorphous composites made of gold and iridium nanoparticles with third-order nonlinear properties embedded in an equally nonlinear host matrix. In the self-phase modulated (SPM) case for so-called Kerr nonlinearities, we give a detailed account on the conditions used to observe saturated and induced absorption, [16] as was discussed in experiments on nanoplanets [17]. Our findings are important for understanding the mechanism of Kerr nonlinearity in amorphous composites and nonlinear enhancement in metallic composites with little losses at low densities. Moreover, our approach can be used to investigate stacks of amorphous composites building nonlinear hyperbolic materials.

We relied on tabulated material data [18,19] for the linear material models. As detailed below, we used a third-order nonlinear susceptibility from the anharmonic oscillator model, ascribing the nonlinear optical response to the resonant excitation of the free electron gas, i.e., collective electron oscillations known as plasmons. To obtain a nonlinear amplitude, we fitted to experimental data available for gold [20] and to the common low frequency result from Miller’s rule for gold and iridium, respectively [21,22]. We found the latter to overestimate the susceptibilities in the resonant regime. The former was in good agreement with a material model based on temporal electron dynamics using the thermo-modulation of the nonlinear response [23].

In nanoparticles, the localized surface plasmon resonance (LSPR) is given by the lowest order electric Mie resonance. In amorphous thin films, this determines the spectral position of their strongest nonlinear response. It is therefore important to take the particle size distribution into account, in particular, for a large size spread [24]. Besides the dipole approximation, the standard Maxwell–Garnett theory [9] uses the quasistatic approximation to the electric polarizability, a term that should be replaced by its corresponding dipolar Mie coefficient for larger particle sizes [24] to correctly include radiation losses. For the thin films that we studied here, we restricted ourselves to metal nanoparticles of 5 nm size, where the dipolar and quasistatic approximation are valid for both discussed materials, gold and iridium. Alumina is a standard material used in the ALD fabrication of amorphous films, and we chose it as the host due to its transparency and stability. In addition, it has an intrinsic nonlinearity [22] of $\chi_{h}^{\left(3\right)}=3.1\times 10^{-22}\;$m${}^{2}$/V${}^{2}$. The small shift in their LSPR changing their diameter from 1 to 20 nm is shown in Figure 1a for Al${}_{2}$O${}_{3}$ as the host material. Figure 1b shows the amorphous structure in mind with metal nanoparticles dispersed in the nonlinear, transparent host material, and Figure 1c illustrates the concept of homogenization within effective medium theories, which are introduced step by step in the next sections. An effective linear permittivity was calculated, as well as an effective nonlinear third-order susceptibility, which, together, form the nonlinear effective permittivity. This latter quantity is dependent on the intensity that the material system is exposed to and, thus, ultimately, on the depth within such an amorphous layer as the intensity drops inside due to absorption.

The next section discusses the foundations of the model used and the assumptions made. We introduce the third-order susceptibility used for gold nanoparticles in Section 2.1 and briefly revisit Miller’s rule in Section 2.2. The focus is then put on the nonlinear Maxwell–Garnett and Bruggeman theories in Section 2.3 and Section 2.4, respectively. Finally, we discuss how we obtained the nonlinear optical properties of amorphous thin films in Section 2.5.

## 2. Methodology

We considered the two metals gold and iridium. While we can rely on their linear bulk properties being well described by tabulated data (gold [18], iridium [19,25]) obtained from ellipsometry measurements, a lack of nonlinear optical data over a wide range of wavelengths makes a simple but consistent model of nonlinear parameters challenging. For a small number of incident wavelengths, pulse durations and measurement techniques, available datasets for gold were collected in [20]. In recent work, similar studies were performed on iridium nanoparticle layers [25]. We used the experimentally available data to obtain a fit to the anharmonic oscillator model amplitude, as well as Miller’s rule [21] for a purely theoretical model for the low-frequency limit ω→0.

### 2.1. Third-Order Nonlinear Susceptibilities

In general, the anharmonic oscillator model [22] for the *i*th component (i∈{x,y,z}) of the third-order macroscopic polarization ${\vec{P}}^{\left(3\right)}$ yields
(1)$$\begin{array}{cc} P_{i}^{\left(3\right)}\left(\omega_{q}\right)= & \epsilon_{0}\sum_{jkl}\sum_{mno}\frac{Nbe^{4}}{3\epsilon_{0}m_{e}^{3}}E_{j}\left(\omega_{m}\right)E_{k}\left(\omega_{n}\right)E_{l}\left(\omega_{o}\right)\delta \left(\omega_{q}-\omega_{m}-\omega_{n}-\omega_{o}\right)\\ \; & \frac{\delta_{ij}\delta_{kl}+\delta_{ik}\delta_{jl}+\delta_{il}\delta_{jk}}{D\left(\omega_{m}+\omega_{n}+\omega_{o}\right)D\left(\omega_{m}\right)D\left(\omega_{n}\right)D\left(\omega_{o}\right)}, \end{array}$$
where *e* is the elementary charge of an electron, $m_{e}$ the electron mass, $\epsilon_{0}$ the free space permittivity, and *N* the electron density, which can be summarized into the plasmon frequency $\omega_{p}$ of the considered material through $\omega_{p}^{2}=Ne^{2}\;/\;\epsilon_{0}m_{e}$. Furthermore, *b* characterizes the strength of the nonlinearity. The first sum runs over the components of the incident electric fields $E_{i}$ at different frequencies. The latter sum over frequencies demands that $\omega_{q}=\omega_{m}+\omega_{n}+\omega_{o}$. This is reflected in the denominators $D\left(\omega \right)=\omega_{0}^{2}-\omega^{2}-i\gamma \omega$, where $\omega_{0}$ is the material resonance stemming from the electron motion and γ is the associated damping due to charge carrier collisions. These are given by the Mie resonances of the respective materials; see Figure 1a. As we are interested in studying amorphous materials, it is important to note that the size distribution should be taken into account if the size range covers multiple resonance frequencies. For the linear regime, this has been addressed by Battie et al. [24]. As can be seen from Figure 1a, for particle sizes below 10 nm, only small variations in the resonance position are expected within classical electrodynamics and, thus, we did not consider a shift in the resonance positions in our thin film calculations. Here, nonlinear properties were studied purely for the case of self-phase modulation, i.e., all fields are monochromatic at the same frequency ±ω, so that a major contribution to the macroscopic polarization stems from the nonlinear response at the incident frequency. The anharmonic oscillator model yields the third-order susceptibility at frequency ω as $\chi^{\left(3\right)}\left(\omega |\omega ,\omega ,-\omega \right)=B\;/\;{\left(|D\left(\omega \right)|D\left(\omega \right)\right)}^{2}$ for materials with an electronic response.

### 2.2. Amplitude According to Miller’s Rule

Miller’s empirical rule [21,22] connects the nonlinear strength *b* to material constants in the limit ω→0. This allows for an assessment of the nonlinear amplitude. Here, we relied on fundamental properties of the solid crystal structure, such as the lattice constant $a_{0}$, which gives an estimate of the displacement that an electron in the atomic crystal structure can experience. The shortest distance between atoms in a face-center cubic (fcc) atomic lattice is $a_{0}\;/\;\sqrt{2}$.

For the third order, it implies that $b=2\omega_{0}^{2}\;/\;a_{0}^{2}$, yielding the low-frequency limit according to Miller’s rule as
(2)$$\begin{array}{c} \chi_{\mathrm{Miller}}^{\left(3\right)}=\chi^{\left(3\right)}\left(\omega \to 0\right)=\frac{2\omega_{p}^{2}e^{2}}{m_{e}^{2}\omega_{0}^{6}a_{0}^{2}}=\mathrm{const}. \end{array}$$

In this limit, 5 nm gold nanoparticles ($a_{0}=4.078$ Å, γ=0.13eV, $\omega_{p}=8.9\;$eV) close to their Mie resonance $\omega_{0}=2.23\;$eV have an estimated third-order nonlinear susceptibility of $\chi_{\mathrm{Miller},\mathrm{Au}}^{\left(3\right)}=2.0\times 10^{-15}\;$m${}^{2}$/V${}^{2}$, whereas this yields $\chi_{\mathrm{Miller},\mathrm{Ir}}^{\left(3\right)}=2.51\times 10^{-17}$m${}^{2}$/V${}^{2}$ for iridium ($a_{0}=3.839$ Å, $\omega_{0}=4.87$ eV, γ=0.18eV, $\omega_{p}=7.2\;$eV) both in Al${}_{2}$O${}_{3}$. Note that the environment enters this estimation through the Mie resonances calculated for the nanoparticles in that specific host material.

The bulk susceptibilities of gold and iridium are compared in Figure 2 for the two cases using Miller’s rule for Au and Ir and experimental data for Au [20] to find a nonlinear amplitude. Miller’s rule can support the understanding of the general behaviour of such systems; however, it shows a strong discrepancy compared to the bulk susceptibility fitted to available experimental values. In general, Miller’s rule overestimates the nonlinearity of metal structures under resonant conditions [26]. A further factor that comes into play during experiments is the pulse duration of the incident light field, which can shift the measured susceptibility by many orders of magnitude as was shown in [20]. The anharmonic oscillator model yields an instantaneous result that corresponds to τ→0 and does not include the aspect of a finite pulse duration. The density functional theory (DFT) could take into account the impact of a finite pulse [27,28,29] and will be discussed in future work. Finally, it should be noted that, in the case of gold, measurements under both resonant and off-resonant conditions are available.

### 2.3. Nonlinear Maxwell–Garnett Theory

In this work, we compared results for nonlinear gold and iridium nanoparticles in alumina for different fill fractions *f* describing the relative volume density of particles in the host. Within the effective medium theory (EMT), amorphous composites are described as a homogeneous film with an effective (linear) permittivity [9] and, in its nonlinear extension, an effective third-order susceptibility as the nanoparticles are assumed to be centrosymmetric and, thus, do not exhibit a second-order nonlinear response.

The Maxwell–Garnett (M-G) theory—see a detailed tutorial for the linear properties [9]—assumes spherical inclusions of a material *i* with permittivity $\epsilon_{i}$ arbitrarily distributed in a host material *h* with permittivity $\epsilon_{h}$. The volume fraction $f=V_{i}\;/\;V_{h}=N4\pi R_{av}^{3}\;/\;3V_{h}$ for *N* inclusions of average size $R_{av}$ has to be small, so that we can exclude interactions between them with distances larger than the wavelength. Moreover, assuming sufficiently small spherical inclusions allows us to work in the dipolar regime and electrostatic approximation [30], where the polarization of a single nanoparticle becomes $\vec{p}=\alpha_{1}^{E}\vec{E}$ with the (electric) polarizability given by
(3)$$\begin{array}{c} \alpha_{1}^{E}=R^{3}\frac{\epsilon_{i}\left(\omega \right)-\epsilon_{h}\left(\omega \right)}{\epsilon_{i}\left(\omega \right)+2\epsilon_{h}\left(\omega \right)}\equiv R^{3}\;\eta \left(\omega \right). \end{array}$$

In this approximation, the effective permittivity is
(4)$$\begin{array}{c} \epsilon_{\mathrm{eff}}=\epsilon_{h}\left(\omega \right)\frac{1+2\eta (\omega )f}{1-\eta (\omega )f}. \end{array}$$

For larger particle sizes or a large particle size variance, the quasistatic polarizability should be replaced by the lowest order Mie coefficient [24] to account for radiation losses and other size effects, such as a size-dependent shift in the resonance position.

The effective third-order susceptibility for nonlinear nanoparticle inclusions $\chi_{i}^{\left(3\right)}\left(\omega \right)$ reads [10,11,12]
(5)$$\begin{array}{cc} \chi_{\mathrm{eff}}^{\left(3\right)} & =f\mu^{2}{\left|\mu \right|}^{2}\chi_{i}^{\left(3\right)}, \end{array}$$
with the well-known enhancement factor from the linear response of a nanoparticle in quasistatic approximation
(6)$$\begin{array}{cc} \mu & =\frac{\epsilon_{\mathrm{eff}}\left(\omega \right)+2\epsilon_{h}\left(\omega \right)}{\epsilon_{i}\left(\omega \right)+2\epsilon_{h}\left(\omega \right)} \end{array}$$
and the effect of a nonlinear host material $\chi_{h}^{\left(3\right)}\left(\omega \right)$ can be included as well, though its impact is rather small with metallic inclusions.

### 2.4. Nonlinear Bruggeman Theory

For high concentrations, i.e., larger volume fractions *f*, and more homogeneous mixtures, the Bruggeman (Br) model for the effective permittivities is preferred [10]. The general equation describing the mixture of two materials is given by
(7)$$\begin{array}{c} f\frac{\epsilon_{i}-\epsilon_{\mathrm{eff}}}{\epsilon_{\mathrm{eff}}+g\left(\epsilon_{i}-\epsilon_{\mathrm{eff}}\right)}+\left(1-f\right)\frac{\epsilon_{h}-\epsilon_{\mathrm{eff}}}{\epsilon_{\mathrm{eff}}+g\left(\epsilon_{h}-\epsilon_{\mathrm{eff}}\right)}=0. \end{array}$$

Note that no difference between a host and an inclusion material is made and, thus, there are no restrictions to the shape of the inclusions. Both constituents are treated on equal footing and the resulting differences to the Maxwell–Garnett theory can be striking [14,31]. The geometrical factor *g* depends on the shape of the inclusions and is typically set to $\frac{1}{3}$ for three-dimensional inclusions (used in this work) and $\frac{1}{2}$ for two-dimensional inclusions. This equation results in two explicit solutions from a quadratic equation, one of which is physical
(8)$$\begin{array}{c} \epsilon_{\mathrm{eff}}=\frac{-c\pm \sqrt{c^{2}+4\left(1-g\right)g\epsilon_{h}\epsilon_{i}}}{2(1-g)}\;\;\mathrm{with}\;\;c=\left(g-\left(1-f\right)\right)\epsilon_{h}+\left(g-f\right)\epsilon_{i}. \end{array}$$

To determine the physical branch, the condition Im(ϵeff(ω))>0 needs to be fulfilled. The effective degenerate third-order susceptibility in the case of weakly nonlinear composites [32] is
$$\begin{array}{cc} \chi_{\mathrm{eff}}^{\left(3\right)} & =\frac{1}{1-f}\left|\frac{\partial \epsilon_{\mathrm{eff}}}{\partial \epsilon_{h}}\right|\left(\frac{\partial \epsilon_{\mathrm{eff}}}{\partial \epsilon_{h}}\right)\chi_{h}^{\left(3\right)}+\frac{1}{f}\left|\frac{\partial \epsilon_{\mathrm{eff}}}{\partial \epsilon_{i}}\right|\left(\frac{\partial \epsilon_{\mathrm{eff}}}{\partial \epsilon_{i}}\right)\chi_{i}^{\left(3\right)}, \end{array}$$
where, in the case of two constituent materials, the derivatives can be determined analytically to read
(9)$$\begin{array}{cc} \frac{\partial \epsilon_{\mathrm{eff}}}{\partial \epsilon_{h}} & =\frac{g+f-1}{2(g-1)}-\frac{\left(1-f-g\right)A\left(\omega \right)-2\left(g-1\right)g\epsilon_{i}}{2\left(g-1\right)\sqrt{A{\left(\omega \right)}^{2}-4\left(g-1\right)g\epsilon_{h}\epsilon_{i}}},\\ \frac{\partial \epsilon_{\mathrm{eff}}}{\partial \epsilon_{i}} & =\frac{g-f}{2(g-1)}-\frac{\left(f-g\right)A\left(\omega \right)-2\left(g-1\right)g\epsilon_{i}}{2\left(g-1\right)\sqrt{A{\left(\omega \right)}^{2}-4\left(g-1\right)g\epsilon_{h}\epsilon_{i}}},\\ \mathrm{with}\;A(\omega ) & =\epsilon_{i}\left(f-g\right)+\epsilon_{h}\left(1-f-g\right). \end{array}$$

The susceptibilities clearly have two separable contributions from the host and from the inclusion material and are easily reduced to the case of only one component being nonlinear. We assumed the host to be nonlinear with constant [22] $\chi_{h}^{\left(3\right)}=3.1\times 10^{-22}$m${}^{2}$/V${}^{2}$, which has little effect on the overall properties of the films at low fill factors. We denoted the absolute values of the prefactors $f\mu^{2}{\left|\mu \right|}^{2}$ to $\chi_{i}^{\left(3\right)}$ in the Maxwell–Garnett EMT and $\frac{1}{f}\left|\frac{\partial \epsilon_{\mathrm{eff}}}{\partial \epsilon_{i}}\right|\left(\frac{\partial \epsilon_{\mathrm{eff}}}{\partial \epsilon_{i}}\right)$, in the Bruggeman EMT, the corresponding nonlinear local field factors. We considered the nonlinear local field factor from the M-G theory in Figure 3a for gold and Figure 3b for iridium and, with it, the expected nonlinear enhancement of the third-order susceptibility $\max\left|\chi_{\mathrm{eff}}^{\left(3\right)}\left(\omega \right)\right|\;/\;\max\left|\chi_{i}^{\left(3\right)}\left(\omega \right)\right|$ of different amorphous composites in Figure 3c with gold and Figure 3d with iridium nanoparticle inclusions compared to the bulk case. Figure 3a underlines that, for gold, strong enhancement factors are expected both for the linear and the nonlinear material parameters, while it can be seen in Figure 3b that iridium yields enhancement factors of around unity. Considering the maximum of the total effective susceptibility for increasing the metal load of the transparent, nonlinear host material in Figure 3c,d, we found an optimum nanoparticle density with a maximum enhancement factor with respect to the bulk case before absorption processes started to dominate the material response. Furthermore, the increase in the nonlinear response from Ir remains moderate, which could be expected already from comparing the Ir results in Figure 1 and Figure 2, which showed minor changes. These findings are, however, contrary to the observations with the Bruggeman theory, where the values are well below unity (and thus not shown in Figure 3a,b). Here, in Figure 3c, the largest relative nonlinear coefficient for amorphous gold/Al${}_{2}$O${}_{3}$ composites is found for smallest fill fractions. With an increasing metal load and absorption, it drops from the beginning. On the other hand, the predictions for Ir follow a similar path.

The real and imaginary part of the effective third-order susceptibilities of the amorphous mixture in the two considered EMTs are shown in Figure 4, varying the fill fraction. The Bruggeman theory in the upper panel and Maxwell–Garnett theory in the lower panel agree on the resonance position at low fill factors, but differ by two to three orders of magnitude in the predicted amplitudes of the amorphous composite. For the M-G theory in the lower panel, a typical redshift of resonances is observed with an increasing fill factor for gold, but this is not yet seen for iridium at low fill factors. Both the linear and nonlinear local field factors yield a strong increase in the effective susceptibilities with respect to their bulk counterparts for gold, but have little impact on iridium, as they are close to unity. This means that amorphous films are expected to show a stronger nonlinear response than bulk materials, which is reflected in the collected experimental data in [20]. The large susceptibilities observed around the resonance for the M-G theory strongly overestimate the nonlinear response of weakly metallic films, as it would allow for strong alterations in the material properties at moderate incident intensities. We will, therefore, concentrate on further evaluating the impact of the effective nonlinear material parameters predicted by the Bruggeman theory.

### 2.5. Nonlinear Amorphous Thin Films

We considered planar thin films with p-polarized incident light. The total optical response, including third-order polarization effects stemming from Kerr nonlinearities at frequency ω and third harmonic generation (THG) at frequency 3ω, can be described through the electric displacement vector in frequency space
(10)$$\begin{array}{cc} \vec{D}\left(\omega ,\vec{r}\right)= & \epsilon_{0}\vec{E}\left(\omega ,\vec{r}\right)+\epsilon_{0}\chi^{\left(1\right)}\left(\omega \right)\vec{E}\left(\omega ,\vec{r}\right)\\ & +\epsilon_{0}\chi^{\left(3\right)}\left(\omega |\omega ,\omega ,-\omega \right){\left|\vec{E}\left(\omega ,\vec{r}\right)\right|}^{2}\vec{E}\left(\omega ,\vec{r}\right)\\ \; & +\epsilon_{0}\chi^{\left(3\right)}\left(3\omega |\omega ,\omega ,\omega \right){\vec{E}}^{3}\left(\omega ,\vec{r}\right)=\epsilon_{0}\epsilon_{\mathrm{NLO}}\vec{E}. \end{array}$$

Note that we have already taken advantage of the fact that spherical metal nanoparticles exhibit a vanishing second-order nonlinear response. The contribution from the THG term is negligible for frequencies around the incident frequency ω, and third-order nonlinear effects can be considered separately.

For thin film calculations, we determined an effective permittivity $\epsilon_{\mathrm{NLO}}$ from linear calculations [11] in a first step
(11)$$\begin{array}{cc} \epsilon_{\mathrm{NLO}}\left(z,\omega \right) & =\epsilon_{\mathrm{eff}}\left(\omega \right)+\chi_{\mathrm{eff}}^{\left(3\right)}\left(\omega \right){\left|\vec{E}\left(z,\omega \right)\right|}^{2}. \end{array}$$
and—as it is dependent on the incident, local field—iterated the input fields until we obtained a self-consistent result exploiting the scattering matrix scheme for several hundreds of subdivisions of the nonlinear amorphous layer.

## 3. Results and Discussion

Within the composite thin films, the incident light is absorbed by the present metal nanoparticles and we overall expect a typical absorption curve within the homogenized layers. The local intensity modifies the local permittivity depending on the nonlinear coefficients and the depth inside the thin film, which was iterated until a stable output field at each position along the layer was found. In the first step, we present in Figure 5 the modified nonlinear permittivity as a function of both the frequency ω of the incident light wave and the varying fill fraction *f* independent from the film thickness, as different depths will be exposed to different light intensities. We concentrated on the case of gold with parameters fitted to experimentally available values [20], and show in Figure 5a the results for Bruggeman and in Figure 5b the corresponding results for the Maxwell–Garnett theory of the nonlinear permittivity. Due to the nature of the effective susceptibility, we observed spectral regions with an increased or decreased total permittivity depending on the sign of the effective susceptibility around the Mie resonance of the considered nanoparticle. Qualitatively, this is the same result for different metallic materials, where the main contribution to the susceptibility in the anharmonic oscillator model stems from the electronic excitation. Increasing the fill factor increases the overall absorption, with a critical value depending on the material, as depicted in Figure 3c for gold, and overall shifts the position of the maxima and minima observed. Here, again, the large local field factors in the M-G theory strongly overestimate the nonlinear response of the amorphous films, leading to the prediction of strong changes in the modified permittivity that are several orders of magnitude larger than Bruggeman. This is in contradiction to observations made on nonlinear properties of thin films, and we restricted our further considerations to the Bruggeman theory.

Note that the signs of the real and imaginary part are opposite in the respective spectral regions close to the resonance frequency. In total, there are four possible sign combinations, as seen in Figure 4: (i) Both real and imaginary parts can be positive, and typically close to but above the resonance frequency; (ii) the real part is negative shortly before reaching the resonance frequency, whereas the imaginary part is still positive; (iii) far from the resonance, the real part is negative below and (iv) positive above the resonance; however, the sign of the imaginary part depends more subtly on the material model due to the local field factors. These combinations can lead to different behaviors inside the film with an increasing intensity, which is depicted in Figure 6 for iridium. As the incident intensity modifies the local permittivity in Equation (Equation 11), in turn, it changes the local intensity that the particles experience. This analytic case based on Miller’s rule and the Bruggeman theory shows in Figure 6a the change in the total intensity $\Delta I=I_{\mathrm{NLO}}-I_{\mathrm{lin}}$ inside a 300 nm thick composite film as a function of the position, varying the fill fraction at ω=4.6 eV. Including forward and backward scattering contributions leads to visible interference effects for low fill fractions, where the absorption is still weak. The changes induced by the nonlinearity are in the range of a few ${\;}^{\mathrm{MW}}\;/\;{\mathrm{cm}}^{2}$. A self-consistent, iterative scheme was used to recalculate the local, saturated intensity after reintroducing the modified nonlinear permittivity several times. This was performed throughout the film to obtain the local permittivity self-consistently with the scattering matrix scheme for planar systems using a few hundred subdivisions of the film. The change in the imaginary part of the modified nonlinear permittivity ΔImϵ=ImϵNLO−Imϵeff is shown as a function of the depth in Figure 6b and as a function of the incident power density in Figure 6c, respectively. For both dependencies, the change induced by the nonlinearity of the amorphous film is of the order of $10^{-3}$ refractive index units tending towards zero further inside the film, where the local intensity drops towards the linear case. Note that, at the chosen frequency well below the Mie resonance of iridium, we observed induced absorption as the imaginary part of the modified total permittivity increases through a positive susceptibility with the intensity without a bound. Due to that, at approximately ${\;}^{\mathrm{TW}}\;/\;{\mathrm{cm}}^{2}$, it would reach unity. Closer to the resonance of the system, this would further increase, reaching unreasonable values under resonant conditions.

As in Figure 6, we demonstrate in Figure 7 cases for saturated and induced absorption in amorphous gold/Al${}_{2}$O${}_{3}$ films. For frequencies well below the Mie resonance of the gold nanoparticles—see Figure 7a–c—the absorption related to the imaginary part of the nonlinear permittivity decreases for an increasing laser intensity with increasing intensity. This effect is known as saturated absorption [16] and is observed here under the same conditions as in Figure 6c, i.e., the effective susceptibility is positive in its imaginary part. In the Bruggeman theory, the effective susceptibility decreases with the fill factor for gold, which is why the strongest change in the imaginary part of the permittivity is seen for smaller fill fractions in contrast to Ir, where the susceptibility does not change as notably with the fill fraction. In contrast, for frequencies much closer to the Mie resonance as depicted in Figure 7d–f, this behavior is reversed because the imaginary part of the effective susceptibility vanishes at the resonance and reverses its sign. The lowest fill factor shows a weaker saturated absorption as before, whereas the higher fill factors with a larger input from the small but negative nonlinear susceptibility enter the regime of induced absorption. This observation is independent of the considered material as it is directly related to the sign of the susceptibility given by the material-dependent Mie resonance and the overall position. It is shifted to lower frequencies for the amorphous structures with increasing fill factors, as was discussed in Figure 3.

We studied the transmitted power flux of ultra-thin amorphous composite Au/Al${}_{2}$O${}_{3}$ layers in Figure 8a as a function of film thickness using the experimentally available data [20]. The difference between linear and nonlinear case is notable in the relative enhancement factor extracted in Figure 8b. For large enough films, the intensity inside the composite layer drops due to absorption entering the linear regime before reaching the glass substrate. Nonlinear intensity enhancement effects take place within the first tenth to hundreds of nanometers in the homogenized film. The enhancement factors are susceptible to the fill fraction reaching a maximum for different film thicknesses. The detriment of metallic absorption to the nonlinear enhancement is reflected in the fact that the larger fill factors show a decreasing nonlinear enhancement compared to smaller particle densities.

The limits of effective medium theories, in particular for large nanoparticle densities and distances that make interaction effects sizable, can be addressed using rigorous methods, such as multiple scattering from nanoparticles using the nonlinear Mie theory [33,34,35], and a comparison with random nanoparticle distributions at increasing fill factors [36], at the cost of a strongly increased computational effort. A further difficulty with homogenization could be waveguiding effects arising in the host material that are suppressed in EMTs. Another interesting aspect is the influence of finite-size effects in ultra-small metal nanoparticles, such as a nonlocal optical response, which can be addressed with semi-classical theories [37,38,39] and ab initio methods [40,41]. This typically yields shifts in the LSPR position, directly impacting the nonlinear susceptibility and, in addition, is accompanied by the nonlocal quenching of the local fields overall reducing local intensities found in the classical picture.

## 4. Conclusions

We studied self-phase modulated nonlinear effects in amorphous composites made of gold and iridium nanoparticles embedded in an equally nonlinear host matrix employing nonlinear formulations of the Maxwell–Garnett and Bruggeman theory. Effective medium theories allow for the homogenization of the linear and nonlinear material properties of amorphous composite layers. The validity is bound to sufficiently low densities, where interparticle effects are negligible both for the linear and the nonlinear regime. For the bulk susceptibility, we considered the anharmonic oscillator model and fitted its nonlinear strength amplitude to both an analytic expression based on Miller’s rule off-resonance and available experimental data for gold susceptibilities. Such homogenized material properties can further be implemented in standard frameworks to evaluate the light interaction with multi-layered systems, such as the scattering matrix theory, reducing the light propagation problem to one dimension. This was used here in a self-consistent approach, i.e., the thin films were subdivided into hundreds of layers and, in each, the modified, depth-dependent nonlinear permittivity was calculated. Moreover, as it is also intensity-dependent, this was carried out several times, recalculating the local intensity that each layer is exposed to when nonlinear enhancement is included. This approach is applicable to nonlinear hyperbolic materials from stacks of amorphous composites and nonlinear enhancement in metallic composites with few losses at low densities. We evaluated thin films of amorphous composites, with a focus on achievable nonlinear enhancement and saturated and induced absorption based on the compositional configuration. Reasonable values were found with the Bruggeman theory using a nonlinear material model fitted to available experimental data for gold. In contrast, both Miller’s rule and the Maxwell–Garnett theory tend to strongly overestimate the nonlinear susceptibility.

## Figures and Tables

**Figure 1 nanomaterials-12-03359-f001:**
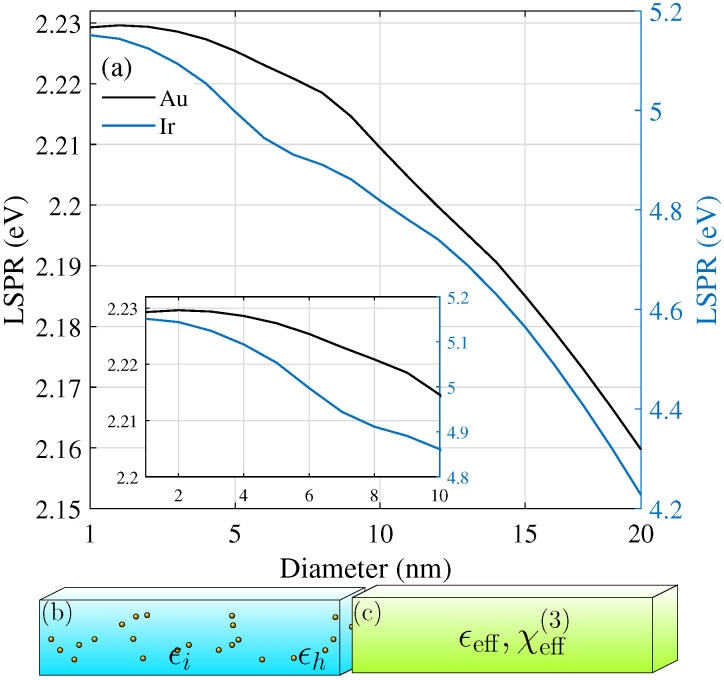
(**a**) Size dependence of the spectral position of the localized surface plasmon resonance (LSPR) of Ir and Au nanoparticles. (**b**) Illustration of the setup: Nanoparticles with permittivity $\epsilon_{i}$ are embedded in the nonlinear host material Al${}_{2}$O${}_{3}$ of permittivity $\epsilon_{h}$. (**c**) The system is treated within the nonlinear Maxwell–Garnett and Bruggeman theories to arrive at a homogeneous layer with effective permittivity $\epsilon_{\mathrm{eff}}$ and effective third-order susceptibility $\chi_{\mathrm{eff}}^{\left(3\right)}$.

**Figure 2 nanomaterials-12-03359-f002:**
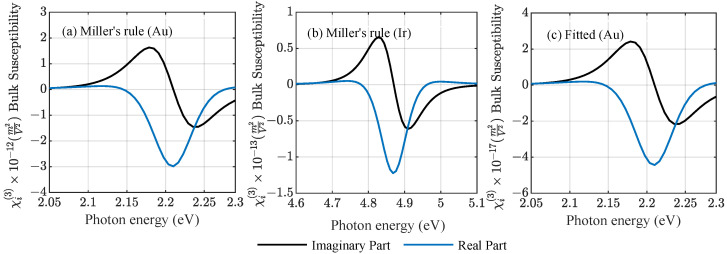
Third-order bulk susceptibility $\chi_{i}^{\left(3\right)}$ as a function of photon energy E=ℏω for (**a**), (**c**) Au and (**b**) Ir using (**a**), (**b**) the analytic Miller’s rule, and (**c**) fitting the nonlinear strength amplitude to available experimental data [20].

**Figure 3 nanomaterials-12-03359-f003:**
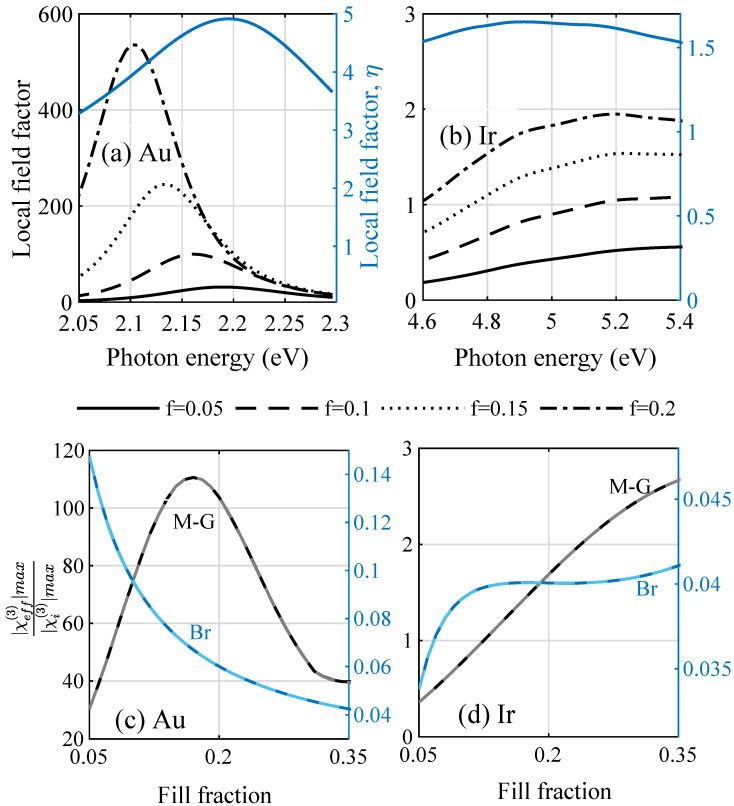
Absolute values of the (**a**,**b**) local field enhancement factors for (**a**) gold and (**b**) iridium in the Maxwell–Garnett theory for the nonlinear coefficients compared to the single particle enhancement factor |η| in the linear case. (**c**,**d**) Enhancement of the nonlinear coefficient comparing the amorphous structures relative to the bulk case for increasing fill factors *f* for both effective medium theories discussed (M-G grey, Br blue), comparing the prediction from Miller’s rule (light color) with the one fitted to experimental values (dark color).

**Figure 4 nanomaterials-12-03359-f004:**
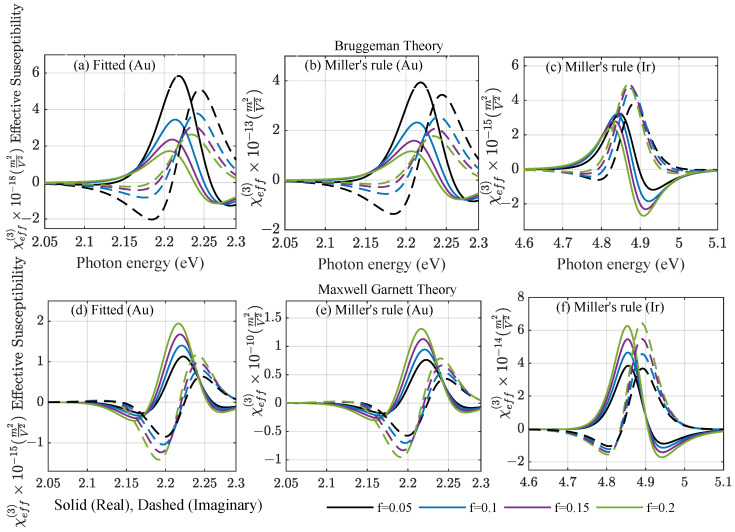
Real (solid) and imaginary (dashed) parts of the third-order effective susceptibility $\chi_{\mathrm{eff}}^{\left(3\right)}$ for nonlinear (upper panel) Br theory and (lower panel) M-G theory as a function of photon energy E=ℏω for (**a**,**b**,**d**,**e**) Au and (**c**,**f**) Ir using (**a**,**d**) fitting the nonlinear strength amplitude to available experiments [20] and (**b**,**c**,**e**,**f**) the analytic Miller’s rule.

**Figure 5 nanomaterials-12-03359-f005:**
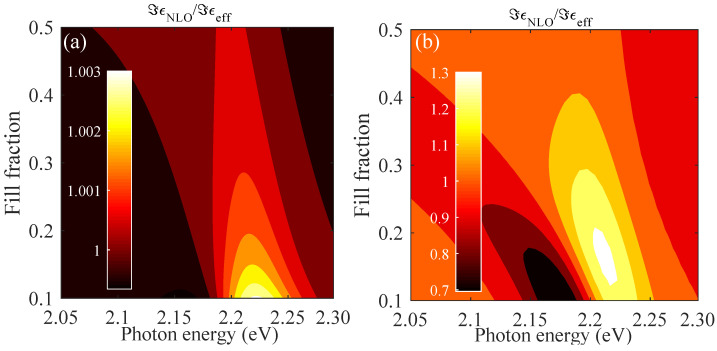
Imaginary part of the modified nonlinear permittivity, Equation (Equation 11), relative to the linear case for gold using the data fitted to available experimental data [20]. Results for (**a**) Bruggeman and (**b**) Maxwell–Garnett theory for varying fill fractions as a function of incident photon energy are compared. The incident field is $|\vec{E}{|}^{2}=7\times 10^{14}\;$V${}^{2}$/m${}^{2}$. Note the strong difference in predicted changes in the local permittivity.

**Figure 6 nanomaterials-12-03359-f006:**
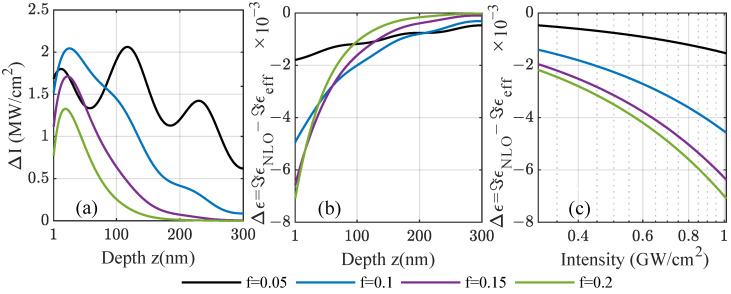
For amorphous composites with varying fill factors of iridium inclusions using the analytic case based on Miller’s rule, the (**a**) change in total intensity (forward and backward scattering contributions) $\Delta I=I_{\mathrm{NLO}}-I_{\mathrm{lin}}$ is shown at incident frequency ω=4.6eV below the related Mie resonance. From the intensity, the change in the imaginary part of the local nonlinear permittivity ΔImϵ=ImϵNLO−Imϵeff is calculated as a function of (**b**) the position inside a 300nm film. The incident power density is $1.6\;{\;}^{\mathrm{GW}}\;/\;{\mathrm{cm}}^{2}$. In (**c**), the change in the modified nonlinear permittivity is studied as a function of incident power, demonstrating induced absorption.

**Figure 7 nanomaterials-12-03359-f007:**
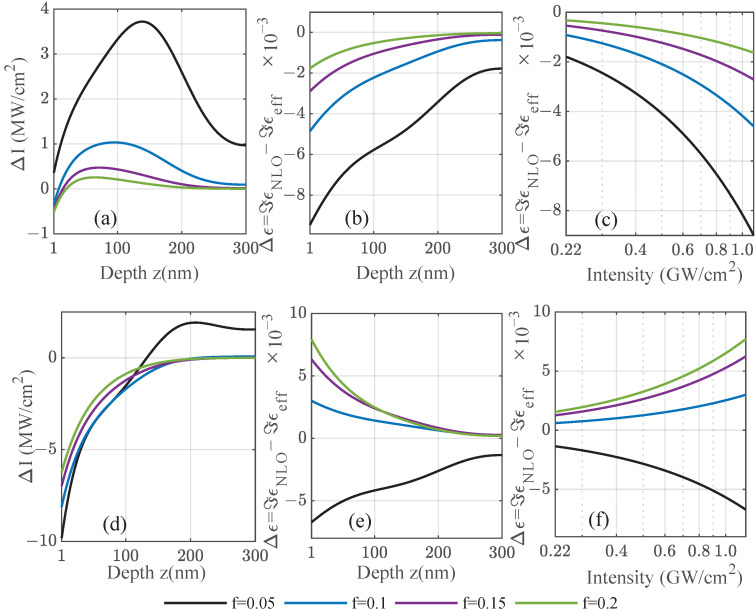
As a function of the position *z* inside a 300 nm composite film for frequencies below the Mie resonance of gold nanoparticles, the (**a**,**d**) change in total intensity $\Delta I=I_{\mathrm{NLO}}-I_{\mathrm{lin}}$ (forward and backward scattering) and (**b**,**c**,**e**,**f**) change in the imaginary part of the nonlinear permittivity ΔImϵ=ImϵNLO−Imϵeff are shown for gold with input values fitted to available experiments [20]. Cases with saturated and induced absorption are observed. The set frequencies are 2.15 eV in the upper panel and 2.20 eV in the lower panel. The incident power density is $1.6\;{\;}^{\mathrm{GW}}\;/\;{\mathrm{cm}}^{2}$.

**Figure 8 nanomaterials-12-03359-f008:**
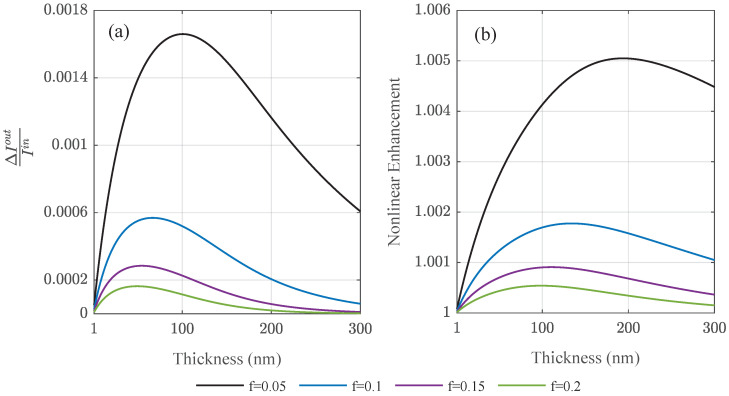
Varying the film thickness of amorphous composite films of different fill fractions *f*, (**a**) the normalized difference in transmitted intensity $\Delta I=I_{\mathrm{NLO}}-I_{\mathrm{lin}}$, and the (**b**) relative intensity enhancement due to nonlinear effects in ultra-thin metallic amorphous films are shown. The incident power density is $1.6\;{\;}^{\mathrm{GW}}\;/\;{\mathrm{cm}}^{2}$.

## Data Availability

The datasets generated and analyzed during the presented study are available upon reasonable request from the corresponding author.

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
