# Peer review of "Thin Films of Nonlinear Metallic Amorphous Composites"

_nanomaterials, 2022, doi:10.3390/nano12193359_

Round 1

Reviewer 1 Report

The  third order Kerr nonlinear response of metallic amorphous composite layers have been studied.  Both saturated and induced absorption in thin films of  amorphous composites are observed depending on the selected frequency and relative position to  the resonant frequency of electron excitation in the metallic inclusions.

The paper is well organized and clearly presented, and the findings are important. Therefore, the referee recommends it to be published. 

Author Response

We would like to thank the reviewer for their positive report. No further action was taken.

Reviewer 2 Report

Dear Authors, could you please comment on the choice of alumina instead of other nonlinear materials?

Reviewer 3 Report

The authors studied self-phase modulated nonlinear effects in amorphous composites made of gold and iridium nanoparticles embedded in an equally nonlinear host matrix employing nonlinear formulations of Maxwell-Garnett and Bruggeman theory. This topic is original and interesting. 

The approach is applicable to nonlinear hyperbolic materials from stacks of amorphous composites and nonlinear enhancement in metallic composites with little losses at low densities, which is novel compared to other published results. 

The computational results are synthetically interpreted and illustrated with appropriate graphics. The paper is well organized and clearly presented, and the findings are important for understanding the mechanism of Kerr nonlinearity in amorphous composites. Therefore, the referee recommends it to be published. 

Author Response

We would like to thank the reviewer for their positive report. The summary reflects well upon our intention. No further action was taken.

Reviewer 4 Report

This manuscript studies the nonlinear optical response of metal amorphous composite layers from the perspective of self phase modulation third-order Kerr nonlinearity. By observing the saturation and induced absorption in the amorphous composite film, it is proved that the film is deeply affected by the nonlinear enhancement effect. There are still lots of questions in the manuscript that need to be revised:

1)     The introduction section was written terribly, the significance of this paper was mentioned vaguely. Also, the structure was also worrisome.

2)     Localized surface plasma resonance (LSPR) is the key experimental direction in the manuscript, and it has not been explained in detail.

3)     In the explanation of Fig. 1b and c, it is said that the concept of effective medium theory is proved, but there is no specific theory or calculation to support it.

4)     The simulation comparison in the manuscript is too scattered without a mainline.  

5)     The conclusion part is too long and needs to be improved.

Round 2

Reviewer 4 Report

All me  concerns have been resolved